# Adding Value to Brewery Industry By-Products as Novel Ingredients in Non-Alcoholic Malt Beverage Applications

**DOI:** 10.3390/foods14162882

**Published:** 2025-08-20

**Authors:** Muhammad Usman Akram, Helen Oluwaseun Agunbiade, Deepak Kadam, Rotimi Emmanuel Aluko, Filiz Koksel

**Affiliations:** 1Food and Human Nutritional Sciences Department, University of Manitoba, Winnipeg, MB R3T 2N2, Canadaagunbiah@myumanitoba.ca (H.O.A.); deepak.kadam@umanitoba.ca (D.K.); 2Richardson Centre for Food Technology and Research, University of Manitoba, Winnipeg, MB R3T 2N2, Canada

**Keywords:** brewer’s spent grain, brewer’s spent yeast, protein hydrolysates, water solubility, novel protein ingredients, non-alcoholic malt beverages

## Abstract

The growing population and increasing concerns about food security and sustainability demand innovative solutions to minimize food waste and transform by-products into functional ingredients valuable to the food sector. Brewery by-products, including brewer’s spent grain (BSG) and brewer’s spent yeast (BSY), are underutilized resources despite their high protein contents and potential as sustainable food ingredients. This study aimed to transform BSG and BSY into protein hydrolysates (BSGH and BSYH, respectively) through enzymatic hydrolysis and thus add value to these brewery industry by-products to be used in the food industry. These protein hydrolysates were incorporated into non-alcoholic malt beverages at three different concentrations, and their effects on the physicochemical properties, including color, kinematic viscosity, turbidity, foaming capacity and foam stability, of the non-alcoholic malt beverages were evaluated. Both BSGH and BSYH exhibited higher water solubility (WS) and lower water binding capacity (WBC) values when compared to their native non-hydrolyzed forms, enhancing their suitability as ideal ingredients for protein supplementation of a wide range of food and beverage products. The production of peptides of varying sizes underscored the effectiveness of enzymatic hydrolysis which resulted in an increase in cysteine and methionine levels in BSYH but a decrease in BSGH. The addition of BSGH and BSYH increased the kinematic viscosity and turbidity but reduced the lightness values in color of the non-alcoholic malt beverages. When the properties of the protein hydrolysates were compared, BSYH was more effective than BSGH in forming foams and maintaining their stability for longer periods. These findings highlight the potential of brewery by-products, after enzymatic hydrolysis, as protein-rich ingredients that can support more sustainable food systems and contribute to the nutritional enhancement of various low-protein food and beverage products.

## 1. Introduction

Barley (*Hordeum vulgare* L.) is the main crop used in the brewing industry, while brewer’s spent grain (BSG) and brewer’s spent yeast (BSY) are the two main by-products of brewing. BSG and BSY account for about 85% and nearly 15% of total brewery wastes, respectively [1]. BSG is the by-product left after the extraction of soluble sugar molecules from malted barley during the beer mashing process. The amount of BSG can reach up to 20 kg after producing 100 hectoliters of beer [2]. BSG is mainly composed of proteins (19–30%), cellulose (12–25%), hemicellulose (20–25%), lignin (12–28%), lipids (10%), and ash (2–5%) on dry weight basis [3]. BSG proteins include hordeins (prolamins), glutelins, globulins, and albumins [4]. On the other hand, BSY is relatively richer in protein (45% dry weight) and can be incorporated into food formulations due to its substantial levels of vitamins, β-glucan, and other fibers [5]. In addition, BSY is generated in considerable quantities, accounting for more than 3 kg per 100 hectoliters of beer [6].

Despite their abundance, BSG and BSY are predominantly used as animal feed after minimal processing and are often overlooked by the food industry. Their limited use in food formulations is largely attributed to their complex composition and high moisture content, which pose challenges in transportation and storage [5,7]. As a result, these by-products are frequently discarded in landfills, where microbial degradation generates methane, contributing to environmental pollution and air quality issues [8].

BSG incorporation into food products, such as granola bars and baked goods, has already been attempted for the purpose of fiber and protein content enrichment [9]. However, its utilization is still very limited due to challenges of achieving desirable sensory acceptance and physical properties, e.g., incorporating high levels of BSG (above 10%) in bread leads to darker color, harder texture, reduced loaf volume, and lower sensory scores for taste, flavor, and overall acceptability [10]. An alternative way to utilize BSG in food and beverage applications is to add value to its proteins through hydrolysis and using the obtained hydrolysates in food and beverage formulations. The proteins in BSG can be hydrolyzed using acidic, alkaline or enzymatic treatments, either individually or in combination. In a study by Vieira et al. [11], alkaline treatment was used to extract protein-rich fractions and arabinoxylans from BSG. However, enzymatic hydrolysis is an effective approach for improving the digestibility and nutritional benefits of BSG by breaking down complex proteins into smaller peptides and free amino acids [7].

In parallel, BSY is composed of protein-rich yeast *Saccharomyces cerevisiae* cells protected by a rigid and thick cell wall. The utilization of protein-rich yeast extracts as human dietary supplements and flavor enhancers was well explained by Oliveira et al. [12]. An investigation was conducted to examine how different concentrations of BSY affect the physicochemical qualities of extruded high-moisture meat analogs. It was discovered that additions of up to 10% were appropriate for obtaining optimum texturization [13]. In another study, Guo et al. [14] demonstrated the possible usage of selenium-rich BSY peptides as natural antioxidants in functional foods and cosmeceutical products.

The thick, rigid cell wall of BSY can be broken down by a variety of mechanisms, including induced autolysis, mechanical rupture, or enzymatic hydrolysis. The latter is also effective in the hydrolysis of complex proteins into smaller peptides [15]. In two different studies, Protamex and Flavourzyme enzymes were utilized to hydrolyze BSG and BSY proteins into smaller peptides and free amino acids, resulting in improved protein solubility [16,17]. In addition, yeast hydrolysates include a considerable proportion of hydrophobic and basic peptides, which exhibit antihypertensive and antioxidant properties [17].

Protein hydrolysates are the set of peptides that are released after the action of food grade protein digesting enzymes. After protein digestion, some of the hydrolyzed peptides can exhibit health benefits including antihypertensive, antioxidant, antidiabetic and anticancer properties [18]. While the biggest appeal of protein hydrolysates for the food industry has so far been on potential bioactivity, their effects on the technological properties (e.g., texture, viscosity) of foods have not been widely studied, which provides a unique opportunity for value-added utilization in a wide range of food and beverage products. Given these gaps, bioactive protein hydrolysates have great potential to emerge as value-added novel ingredients in these sectors. Hydrolyzed proteins can be extracted from BSG and BSY in the form of BSGH and BSYH, respectively. These protein hydrolysates can then be used in the formulation of functional food and beverage products. There is limited information on the application of brewery-based protein hydrolysates in the development of protein rich non-alcoholic malt beverages. Overall, transforming these low-cost industrial by-products into valuable food components offers several benefits by recycling materials that might otherwise contribute to landfill waste. Additionally, this transformation reduces the risk of environmental pollution and fungal contamination from improper storage.

The aims of this study were (1) to produce protein hydrolysates from BSG and BSY through enzymatic hydrolysis and to characterize them, and (2) to incorporate these hydrolysates into non-alcoholic malt beverages and assess their effects on important beverage physicochemical properties, i.e., color, viscosity, turbidity, and foaming properties, thereby evaluating the potential for enhancing the value and functionality of beverages. This research can potentially support the development of innovative approaches that will enhance the sustainability and benefits of the brewing industry from an economical perspective by adding value to its by-products.

## 2. Materials and Methods

### 2.1. Materials

BSG and BSY samples, along with non-alcoholic malt beverage (commercial name: Farmery premium non-alcoholic beer having composition %: carbohydrate 4.2, protein 0.1, Fat 0.0, Fiber 0.0) were obtained locally in Manitoba, Canada. The bovine chymotrypsin enzyme was purchased from Sigma-Aldrich (St. Louis, MO, USA). All chemicals were of analytical grade with high purity.

### 2.2. Proximate Composition

Moisture, fat, and ash contents of the BSG and BSY were determined according to AACC Method No. 44-01, 30-20 and 08-01, respectively [19]. Protein content of BSG and BSY were determined using the modified Lowry method [20]. However, the total carbohydrates were calculated by subtracting the above values from 100.

### 2.3. Production of Protein Hydrolysates

The protein hydrolysates were produced from BSG and BSY through enzymatic hydrolysis, according to the method of Asen and Aluko [21], with slight modifications. Preliminary analysis indicated that chymotrypsin was the most effective enzyme for releasing peptides from BSG and BSY, and a hydrolysis time of 4 h provided an optimal balance between peptide yield, functional properties, and enzyme efficiency. Papain was also used for hydrolysis under its optimal pH and temperature conditions (pH 6.5, 60 °C) for the same duration, but it showed significantly lower peptide yield compared to chymotrypsin. Briefly, dried samples of BSG and BSY were ground using a small blender and passed through a 250 µm sieve. The ground BSG was dispersed in distilled water (based on the 5% protein content of the final solution, *w/v*). The dispersion was adjusted to pH 8.0 and temperature of 37 °C for optimum activity of the chymotrypsin enzyme. Subsequently, chymotrypsin was added to the dispersion as 1% (*w/w*) of the total raw material, followed by continuous mixing on a magnetic stirrer. After 4 h of hydrolysis, the reaction was stopped by inactivation of the enzyme through adjustment to pH 4.5 using 2 N HCl and heating the mixture for 15 min at 95 °C. The digest mixture was then cooled to room temperature and centrifuged at 1460× *g* at 4 °C for 30 min. The supernatant was freeze-dried, ground, and then stored in fine powder form at −18 °C as BSG protein hydrolysate (BSGH). The same procedure was followed to produce BSY-based protein hydrolysate (BSYH) from BSY (Figure 1).

The moisture, fat, ash, and carbohydrate contents of the BSGH and BSYH were measured in triplicate using the same methods previously described for BSG and BSY. The Dumas combustion method was employed for measuring the protein content of BSGH and BSYH in triplicate using an Elementar rapid N exceed protein analyzer (Langenselbold, Germany). Conversion factors of 5.83 for barley [22] and 5.80 for residual brewer’s yeast [23] proteins were used.

### 2.4. Water Solubility (WS) and Water Binding Capacity (WBC)

Water solubility (WS) and water binding capacity (WBC) of the BSGH and BSYH were measured according to the method reported by Li et al. [24]. Briefly, 0.5 g of BSGH (or BSYH) was transferred to 5 mL of distilled water in a 15 mL falcon tube. The mixture was vortexed for 15 s every 5 min for 1 h and then centrifuged at 5000× *g* for 10 min at room temperature. After carefully transferring the supernatant to a moisture dish and weighing the residual precipitate, both the supernatant and the moisture dish were placed in an oven (Heratherm OGS100, Thermo Scientific, Braunschweig, Germany) for overnight drying at 100 °C.

WS (%) and WBC (%) were determined in triplicate using the following equations:
$$WS=\frac{Weight\;of\;dried\;supernatant}{Weight\;of\;protein\;hydrolysate}\times 100\;WBC=\frac{Weight\;of\;wet\;precipitate-Weight\;of\;dried\;precipitate}{Weight\;of\;protein\;hydrolysate}\times 100$$

### 2.5. Emulsion Capacity (EC)

Emulsion capacity (EC) of the BSGH and BSYH was measured according to Luo and Koksel [25], with minor modifications. Briefly, 0.5 g of BSGH (or BSYH) was dissolved in 5 mL of distilled water in a 15 mL falcon tube and mixed for 30 min. After the addition of 3 mL of canola oil, the obtained mixture was homogenized (IKA T-18 basic Ultra-Turrax, Konigswinter, Germany) at 24,000 rpm for 2 min and centrifuged at 3000× *g* for 20 min at room temperature. EC (%) was calculated in triplicate using the following equation:
$$EC=\frac{Height\;of\;the\;emulsified\;layer\;in\;the\;tube}{Height\;of\;the\;whole\;mixture\;in\;the\;tube}\times 100$$

### 2.6. Amino Acid Composition and Size Exclusion Chromatography

The amino acid composition, excluding cysteine, methionine, and tryptophan, was determined by hydrolyzing the samples for 24 h with 6 N HCl in accordance with the AOAC official method 982.30 [26]. For cysteine and methionine analysis, protein oxidation was conducted with performic acid prior to hydrolysis, as described in AOAC official method 982.28 [26]. Following hydrolysis, all amino acids were derivatized and separated utilizing a AccQ-Tag C18 column (100 mm × 2.1 mm, 1.7 µm) equipped with a SIL-30 AC autosampler on a Shimadzu UPLC system (Nexera, Kyoto, Japan). The column temperature was maintained at 51 °C for the amino acids except cysteine and methionine, which were detected at 40 °C and 60 °C, respectively [27]. Finally, tryptophan was determined in accordance with ISO standard 13904 following the alkaline hydrolysis of the samples [28]. NIST soy flour standard reference material 3234 was employed for quality control, and hydrated molecular weights of amino acids were utilized to calculate quantities, with results reported as percentages.

The size exclusion chromatography analysis, both before and after enzymatic hydrolysis treatments, was performed using Fast Protein Liquid Chromatography (FPLC). Hydrolysate preparations (20 mg/mL) were dissolved in 0.05 M sodium phosphate buffer containing 0.15 M NaCl (pH 7.2) and then filtered through a 0.2 µm syringe filter. A 100 µL aliquot of the filtrate was injected onto the Superdex 75 10/300 GL column and isocratically eluted at a flow rate of 0.5 mL/min [29]. The analysis was conducted using the AKTA FPLC system (GE Healthcare, Montreal, PQ, Canada). The molecular weight of each peptide peak was determined by correlating the sample’s elution volume to a standard calibration curve. This curve was constructed by plotting the logarithm of molecular weight against the elution volume of reference proteins, including Aprotinin (6.50 kDa), Ribonuclease A (13.7 kDa), Carbonic anhydrase (29 kDa), Ovalbumin (44 kDa) and Conalbumin (75 kDa).

### 2.7. Preparation of the Beverages

Protein-enriched non-alcoholic malt beverages were prepared by incorporating varying amounts of BSYH and BSGH into the commercially available non-alcoholic malt beverages obtained locally in Manitoba (Canada). Three different treatments, labeled as treatment 1, 2 and 3, were developed, in which 2.5 g, 5 g and 10 g of BSGH (or BSYH) (wet basis), respectively, were added to 473 mL (i.e., standard can size) of non-alcoholic malt beverages and mixed for 10 min using a magnetic stirrer. Along with this, a control non-alcoholic malt beverage was used without the incorporation of any protein hydrolysates. For the protein hydrolysate containing beverages, homogenization was carried out using a probe homogenizer (IKA T 18 basic Ultra-Turrax, Konigswinter, Germany) at 7000 rpm for 30 s. Finally, all beverages, including the control, were pasteurized at 60 °C for 15 min and then stored at 4 °C for further analyses.

### 2.8. Color Analyses of the Beverages

Two different color measurement methods were employed, namely the ASBC and the CIE (L*, a*, b*) methods. The ASBC (American Society of Brewing Chemists) color analysis was carried out according to Li and Maurice [30], with minor modifications. Briefly, the absorbance value of a 3 mL aliquot of each beverage sample was measured in a 1 cm cuvette using a UV-visible spectrophotometer (Genesys 150, Thermo Fisher Scientific, Madison, WI, USA) at a wavelength of 430 nm. Distilled water was used as a blank. The beverage color was then calculated by multiplying the absorption value by a factor of 12.7. All results were reported as mean values of triplicate measurements.

The CIELab color analysis was carried out for both (BSG and BSY) in their raw form, as well as their respective protein hydrolysates (BSGH and BSYH), and the non-alcoholic BSGH and BSYH incorporated malt beverages using a colorimeter (Model: CM-3500d, Minolta, Osaka, Japan) at an angle of D65°, according to Koksel and Masatcioglu [31]. Here, L*, a*, and b* indicate lightness–darkness, greenness–redness, and blueness–yellowness, respectively. Chroma (C) indicates color saturation and hue angle (h) describes color tone in 3D color diagrams. Color measurement for each sample was conducted in triplicate using equal amount of the same sample, and mean values were reported.

### 2.9. Viscosity of the Beverages

An Ubbelohde viscometer (1C, Cannon Instrument Company, State College, PA, USA) was used to measure the kinematic viscosity of the beverage samples. Each beverage sample was transferred to a capillary tube, and the efflux time (in seconds) required to pass from the upper mark to the lower mark of the viscometer was measured in triplicate. Kinematic viscosity in centistokes (cSt) was calculated by multiplying the efflux time with the viscometer constant (0.03).

### 2.10. Turbidity of the Beverages

Turbidity analysis was performed following the method described by Sae-Leaw et al. [32], measuring the absorbance value of a 3 mL aliquot in a spectrophotometer (Genesys 150, Thermo Fisher Scientific, Madison, WI, USA) at a wavelength of 660 nm. Distilled water was used as blank. Results were reported as the means of triplicate measurements.

### 2.11. Foaming Capacity and Foam Stability of the Beverages

Foaming capacity and foam stability analyses were performed according to the method described by Aluko et al. [33], with minor modifications. Briefly, 10 mL of beverage samples at pH 4.53 were transferred into 50 mL graduated centrifugal tubes. The samples were then homogenized (IKA T 18 basic Ultra-Turrax, Konigswinter, Germany) at 20,000 rpm for 60 s. The height (mm) of the foam produced was carefully measured. The foam height that remained stable after 30 min at room temperature was expressed as a percentage of the original foam height produced. Results for foaming capacity and foam stability were reported as the means of triplicate analyses.

### 2.12. Statistical Analysis

The statistical analysis was performed using the IBM SPSS Statistics package (version 23.0, IBM Corporation, New York, NY, USA). A one-way analysis of variance (ANOVA) was performed, followed by Tukey’s post hoc test, to examine the statistically significant variation among the means. The statistical variation in results for BSGH and BSYH incorporated non-alcoholic malt beverages in comparison to the control beverage is represented by lower- and upper-case letters, respectively.

## 3. Results and Discussion

The proximate composition of raw materials and hydrolysates in g/100 on dry basis (db) is provided in Table 1. The protein contents of the BSGH and BSYH were relatively improved when compared to the BSG and BSY, respectively. The increase in protein content of hydrolysates is mainly attributed to the removal of non-soluble fractions (such as fat, fibers, and proteins of high molecular weights) during the preparation stage, more specifically the centrifugation step (Figure 1). Our findings are consistent with the values reported in the previously literature [34,35].

### 3.1. Water Solubility, Water Binding Capacity and Emulsion Capacity

Water solubility (WS) is an important parameter for determining the applicability of protein hydrolysates in beverage products. In this study, both BSGH and BSYH exhibited high WS values, as presented in Table 2. The improvement in WS of BSGH and BSYH as compared to BSG and BSY, respectively, can be attributed to the enzymatic hydrolysis step during the production of protein hydrolysates. The enzymatic hydrolysis results in the breakdown of proteins into smaller peptides, releasing new amino- and carboxyl-terminal side chains and therefore increasing ionizable groups [36,37]. These small peptides are more hydrophilic and able to form new hydrogen bonds with water molecules [38]. The high WS of both protein hydrolysates indicates that they contain a higher concentration of soluble proteins with less aggregation compared to their respective raw materials [39]. Similarly to our findings, Abeynayake et al. [40] also reported an increase in the solubility of BSG proteins, reaching up to 94.4%, following treatment with endogenous barley protease enzymes. In another study, soy protein hydrolysate exhibited markedly higher solubility (97.1 g/100 g) than soy protein isolate (33.9 g/100 g) following hydrolysis with the enzyme alcalase. This enhancement was linked to increases in smaller peptides, as well as ionizable carboxyl and amino groups, which improve water solubility [41].

Water binding capacity (WBC) refers to the highest amount of water that can be absorbed or retained by an ingredient when exposed to an external force. It is affected by various factors, such as structure of proteins, presence of ionic salts, pH of solution, and temperature conditions [42]. On the other hand, emulsion capacity (EC) measures the amount of oil that can be emulsified per gram of protein before the emulsion undergoes a phase transition [43]. It mainly depends on the interactions between proteins and oil droplets, which is significantly improved by the abundance of hydrophobic peptides [44].

BSG proteins, such as hordein and glutelin, primarily function as storage proteins with high surface hydrophobicity [34], while BSY has a more complex composition, including structural proteins, signaling proteins, and mannoproteins that contribute to its hydrophilicity [45]. Enzymatic hydrolysis by chymotrypsin results in cleavage of proteins into small peptides, exposing both hydrophobic and hydrophilic sites. This exposure reduced the WBC of BSGH and BSYH. However, the EC values of BSGH and BSYH remained comparable to those of the native BSG and BSY, respectively. However, the WBC and EC values of BSYH were considerably higher than those of BSGH, as presented in Table 2. This difference may be due to how enzymatic hydrolysis influences the surface hydrophobicity in the peptides of each hydrolysate, which directly impacts the WBC and EC values. In a study, Chin, Keppler, Dinani, Chen and Boom [34] observed a decline in surface hydrophobicity of barley protein isolates after pH-adjusted enzymatic extraction. However, Marson, de Castro, Machado, da Silva Zandonadi, Barros, Maróstica Júnior, Sussulini and Hubinger [35] reported an increase in hydrophobicity of BSYH prepared from different proteolytic enzymes. These variations in surface hydrophobicity might be the reason behind differing WBC values of BSGH and BSYH. Overall, enzymatic hydrolysis can improve the techno-functional properties of protein hydrolysates [46] and offers new opportunities for the development of a wide range of protein-rich products including beverages. Beyond these functional improvements, the peptides in BSGH and BSYH are known for their bioactivity, namely antioxidant, antihypertensive, and anti-inflammatory properties [18].

### 3.2. Amino Acid Composition and Size Exclusion Chromatography Analyses

The amino acid profiles of the BSG and BSY, both before and after hydrolysis with chymotrypsin, are presented in Table 3. A total of 18 amino acids were identified across all samples, comprising 11 essential and 7 non-essential amino acids. Glutamic acid and glutamine were the most abundant in all samples, indicating potential functional benefits in food applications consistent with glutamic acid’s role in imparting umami flavor [3]. The hydrolysis of both BSG and BSY by chymotrypsin appeared to have a positive impact on increasing the levels of aromatic amino acids, such as tryptophan, tyrosine, and phenylalanine. This might have happened due to the chymotrypsin enzyme specifically cleaving the peptide bonds at sites containing these aromatic amino acids [47]. The concentrations of sulfur-containing amino acids, such as cysteine and methionine, increased in BSYH but decreased in BSGH, whereas lysine demonstrated a slight reduction in both hydrolysates (BSGH and BSYH) relative to their respective unhydrolyzed forms (BSG and BSY). Both BSG and its hydrolysate BSGH had higher proline, but lower aspartic acid as compared to BSY and its hydrolysate BSYH. Additionally, the hydrolysis of BSY resulted in a reduction in arginine, serine, and alanine levels but an increase in proline. Overall, the major amino acids in BSG, BSY, and their hydrolysates (BSGH and BSYH) were glutamic acid + glutamine, aspartic acid + asparagine, proline, leucine, valine, and phenylalanine, which are also consistent with findings reported in previous studies [16,48].

The size exclusion chromatography (SEC) analysis of BSG, BSGH, BSY, and BSYH revealed distinct peptide molecular weight distributions, highlighting the effects of enzymatic hydrolysis on protein breakdown (Figure 2). BSG displayed two primary peaks at 2.66 kDa and 0.55 kDa, indicating the presence of intermediate and low-molecular-weight peptides. In contrast, BSGH showed a broader profile with peaks at 3.19 kDa, 1.5 kDa, and 0.30 kDa, suggesting the generation of both larger peptides and smaller oligopeptides through hydrolysis. Similarly, BSY exhibited three peaks at 2.76 kDa, 1.43 kDa, and 0.28 kDa, reflecting a simpler peptide profile predominantly comprising low-molecular-weight components relative to BSYH which demonstrated a broader range of peptide sizes with peaks at 3.07 kDa, 1.48 kDa, and 0.26 kDa, indicative of a diverse range of peptides formed during hydrolysis.

Comparing the molecular weight distribution of peptides in hydrolysates (BSGH and BSYH) to their respective unhydrolyzed proteins (BSG and BSY) underscores the effectiveness of enzymatic hydrolysis in generating peptides of varying sizes. Hydrolysates exhibited higher molecular weight fractions (e.g., 3.19 kDa in BSGH and 3.07 kDa in BSYH), likely due to partial hydrolysis or peptide aggregation, along with significant low-molecular-weight fractions (<0.3 kDa), as similarly reported by Abeynayake, Zhang, Yang and Chen [40] for the bioactive peptides of BSG proteins. Similarly, enzymatic hydrolysis by different protease enzymes led to the degradation of fish proteins, namely those from blue whiting, into smaller peptides, with more than 50% of blue whiting protein hydrolysate powders comprising peptides ≤0.5 kDa, facilitating digestion and absorption rates [49]. The smaller peptides are of particular interest due to their potential bioactivity and functional properties [50]. Another study by Zhang et al. [51] on soy protein hydrolysates showed that enzymatic hydrolysis can lead to the formation of large insoluble aggregates, primarily due to hydrophobic interactions between peptides. The broader size distribution in hydrolysates suggests enhanced versatility for applications in food and nutraceuticals [52]. Further investigation into the bioactivity and structural characteristics of these peptides would provide deeper insights into their potential health benefits and functional applications.

### 3.3. Color Analyses

Color is one of the most essential aspects of a beverage, indicating the color properties of the raw materials it is made from, the nature of the processing operations it underwent, and adding to its appeal. Table 4 presents the color properties of the BSG and BSY, and their protein hydrolysates BSGH and BSYH. The color properties of the BSGH and BSYH incorporated beverages are shown in Table 5. Enzymatic hydrolysis led to considerable differences in color, with BSGH becoming lighter and BSYH becoming darker compared to their non-hydrolyzed forms. The total color difference (ΔE) values increased for both BSGH and BSYH compared to the dried BSG and BSY samples, respectively. Additionally, BSGH exhibited an increase in lightness (L*) values, resulting in a lighter appearance, while BSYH showed a decrease in lightness values, causing it to appear darker.

The incorporation of BSGH and BSYH led to a reduction in the lightness (L*) and hue values while a significant (*p* < 0.05) increase was observed in the redness (a*), yellowness (b*), chroma (C), and total color difference (ΔE) values of the prepared non-alcoholic malt beverages. A similar reduction in the L* values was observed in non-alcoholic malt beverages supplemented with BSGH and BSYH at concentrations of 2.5 g and 5 g per 473 mL beverage (treatment 1 and 2, respectively). However, in treatment 3 (10 g of hydrolysate per 473 mL beverage), the lightness of the BSYH incorporated beverages decreased slightly more than that of BSGH incorporated beverages. Additionally, all treatments of BSYH incorporated beverages showed significant increases in yellowness and redness was observed when compared to BSGH incorporated beverages.

The b* values (indicating yellowness) were higher than the a* values (indicating redness), resulting in a distinctly more yellowish shading in all the beverages. Furthermore, the addition of both protein hydrolysates increased the chroma values, leading to higher color saturation, while decreasing the hue angles and contributing to a more yellowish overall appearance. In a previous study, yellowish red or brown color of popped rice was associated with the increase in chroma and decrease in hue values [53]. These variations in color of protein hydrolysate incorporated non-alcoholic malt beverages may be attributed to the original brownish color of the BSGH and BSYH (see the color properties of BSGH and BSYH in Table 4). Additionally, both hydrolysates contain high concentrations of small peptides and free amino acids. During the pasteurization of non-alcoholic malt beverages at 60 °C for 15 min, these free amino acids can undergo a Maillard reaction with reducing sugars present in the beverage. This reaction may result in the formation of brown pigments, leading to the darkening of the beverages [15]. Similarly, Bazsefidpar et al. [54] observed that adding BSG protein hydrolysates to muffins altered crust and crumb color, with shifts in L*, a*, and b* values attributed to Maillard reactions and melanoidin formation.

### 3.4. Kinematic Viscosity

Kinematic viscosity, the internal resistance of a fluid to flow when it is subjected to gravitational forces, is an important characteristic of beverages, affecting the processing conditions and mouthfeel of the processed beverage [55]. Figure 3 demonstrates the effects of varying concentrations of BSGH and BSYH on the kinematic viscosity of the non-alcoholic malt beverages. The addition of BSGH and BSYH increased the kinematic viscosity of the beverages from 1.30 cSt for the control to maximum values of 1.53 cSt and 1.79 cSt for the BSGH and BSYH incorporated beverages, respectively (*p* < 0.05).

Overall, the kinematic viscosity increase was more pronounced for BSYH treatments when compared to BSGH treatments. This rise in kinematic viscosity of the BSYH incorporated beverages can be attributed to the higher WBC values of BSYH compared to that of BSGH. Similarly to our findings, enzymatic hydrolysis combined with ultrasound as a pre-treatment method reduced the particle size of cricket proteins. This size reduction improved the functional properties, such as solubility, gelling capacity, and water-holding capacity of cricket proteins, which ultimately influenced the viscosity of the beverage upon the addition of the smaller-sized cricket proteins [56].

### 3.5. Turbidity

Turbidity (haze) is defined as the level of cloudiness in a liquid due to suspended particles. As expected, both the turbidity and color values of non-alcoholic malt beverages were influenced by the solubility of the raw components and processing conditions, as previously reported by Lozano [57]. This is important because turbidity not only influences the visual appeal of beverages but also affects the perception of quality by consumers. The level of turbidity may be either desirable or undesirable depending on the type of product and preference of consumers [58].

The addition of BSGH and BSYH increased the turbidity of the beverages, with BSYH showing lower turbidity values compared to BSGH, as shown in Figure 4. The results indicated that turbidity values increased by 200%, 287%, and 412% relative to control (0.08) with increasing concentrations of BSGH. Similarly, the addition of BSYH led to an increase in turbidity values from 137% to 375%. The addition of BSGH led to a more pronounced increase in the turbidity of the beverages than incorporating the BSYH. In contrast, BSYH contributed more significantly to viscosity enhancement of the beverages compared to BSGH. The increased turbidity in BSGH incorporated beverages may be primarily due to the presence of flavanol polyphenols and proline-rich polypeptides, which mainly originate from the hordein protein in barley grains [59].

### 3.6. Foaming Capacity and Foam Stability

Protein structures possess amphipathic properties, with both hydrophilic and hydrophobic surfaces. This characteristic facilitates foam formation at air–liquid interfaces, prevents bubble coalescence, and helps stabilize the foam [60]. Enzymatic hydrolysis breaks down and unfolds specific structures of proteins, revealing hydrophobic regions within peptides, letting them diffuse more quickly at air–liquid interfaces and make elastic films around the air bubbles, influencing the foaming capacity [61].

Foaming capacity refers to the ability of proteins to rapidly adsorb at the air–water interface during whipping or aeration, reducing surface tension and enabling the formation of a large volume of foam [62]. By modifying insoluble proteins, hydrolysis improves foaming properties through more efficient adsorption and diffusion at air–liquid interfaces. The addition of BSYH improved the foaming capacity of non-alcoholic malt beverages, as shown in Figure 5a. The addition of BSGH produced foaming capacity similar to or less than that of the control. On the other hand, incorporating BSYH led to a significant increase in foaming capacity, reaching 115%, 120%, and 119%, respectively, when compared to the control beverage, as the concentration of BSYH increased. By modifying insoluble proteins, hydrolysis improves foaming properties through more efficient diffusion of peptides to air–liquid interfaces in foams and their adsorption at the interfaces. In another study, soy protein hydrolysate demonstrated markedly higher foaming capacity (~22%) and stability than soy protein isolate (~13%) following alcalase hydrolysis [41].

Foam stability is the ability of proteins to stabilize the produced foam against various stress factors. It mainly depends on the mechanical strength of interfacial protein films around the air bubbles [63]. The addition of BSGH was unable to form a stable foam in BSGH incorporated beverages, whereas improved foam stability was only observed in BSYH incorporated beverages (Figure 5b). BSYH addition resulted in a significant (*p* < 0.05) improvement in foam stability, with values of 29.55%, 44.85%, and 53.66%, as its concentration increased when compared to the control beverage of 21.75%. Similarly to our results, Vieira et al. [64] did not observe any significant effect of hydrolysis on the foaming capacity or foam stability of BSG protein hydrolysate (BSGH). This reduction in foam stability of BSGH-added beverages may be due to extensive hydrolysis resulting in an abundance of small-sized peptides compared to native protein and less amphiphilic peptides with a reduced ability to form stable foams [61]. The pH values of the control non-alcoholic malt beverages were ~4.5, which explains the reduction in foam stability in BSGH incorporated non-alcoholic malt beverages. The pH-dependent nature of barley protein is another important factor contributing to the reduction in foaming capacity, showing higher foaming properties at alkaline pH values and reduced foam stability at the acidic pH values of the beverages [65,66].

The valorization of BSG and BSY into functional protein hydrolysates aligns with waste reduction and circular economy goals [67]. Partially or fully replacing dairy or soy proteins with BSGH or BSYH can reduce agricultural land and water use, supporting sustainable consumption and climate action targets. Moreover, partnerships between breweries and beverage manufacturers could foster closed-loop bioeconomy networks, strengthening both environmental and economic performance [68]. However, the successful market adoption of such upcycled or fortified beverages ultimately depends on consumer acceptance, which requires balancing perceived health benefits with appealing sensory qualities. While sustainability claims may increase willingness to pay, purchasing decisions are still driven largely by flavor, color, and mouthfeel [69]. Barriers to adoption include skepticism toward waste-derived ingredients and concerns about unfamiliar tastes [70]. To overcome these challenges, strategies such as clear labeling, evidence-based health claims, and gradual introduction of functional levels can be effective. Given the limited research on consumer perceptions of high-protein non-alcoholic malt beverages, future studies on sensory quality and market adoption are warranted.

## 4. Conclusions

This study aimed to transform two by-products of the brewery industry, BSG and BSY, into protein hydrolysates through enzymatic hydrolysis, producing novel and small-sized soluble peptides. Our findings revealed that the incorporation of BSGH and BSYH into non-alcoholic malt beverages improved the protein content and enhanced important physicochemical properties such as kinematic viscosity, turbidity, and foaming capacity, when compared to the control beverage. Foam stability was significantly improved only with BSYH addition, whereas BSGH had no measurable effect on foam stability. BSGH incorporation primarily affected turbidity, while BSYH had a more significant impact on color, viscosity, and foaming properties of the non-alcoholic malt beverages. The addition of both hydrolysates slightly enhanced the dark appearance of the beverages, as indicated by an increase in lightness and chroma values. This study investigated the effect of BSGH and BSYH incorporation on the physicochemical properties of the non-alcoholic malt beverages at various concentrations; however, a sensory evaluation is necessary to determine the optimal concentration for consumer acceptance. Among all tested formulations, BSYH at 2.5 g/473 mL provided the closest match to the control beverage in terms of color, viscosity, and turbidity, while also significantly improving foaming capacity and stability. Therefore, BSYH at low concentration appears to be the optimal choice for fortifying non-alcoholic malt beverages with minimal deviation from the original product characteristics.

Overall, the high water solubility of the hydrolysates produced using chymotrypsin enzyme supports broad applications of these hydrolysates in both alcoholic and non-alcoholic products. This study addresses a critical knowledge gap by offering the first comprehensive compositional and techno-functional assessment of protein hydrolysates derived from brewery by-products for use in the beverage industry. The results demonstrate that proteins in these by-products can be transformed into application-ready functional peptides, advancing sustainable protein innovation and serving as a practical model for other agri-food sectors pursuing circular economy principles. Large-scale adoption of the results will require addressing challenges such as enzyme cost, energy needs for drying and storage, variability in raw material quality, and sensory issues including bitterness and color changes from Maillard reactions, along with potential regulatory requirements in certain markets. Future studies should optimize nutritional benefits, conduct sensory evaluations, and assess shelf-life stability, with particular attention to precipitation risks during storage.

## Figures and Tables

**Figure 1 foods-14-02882-f001:**
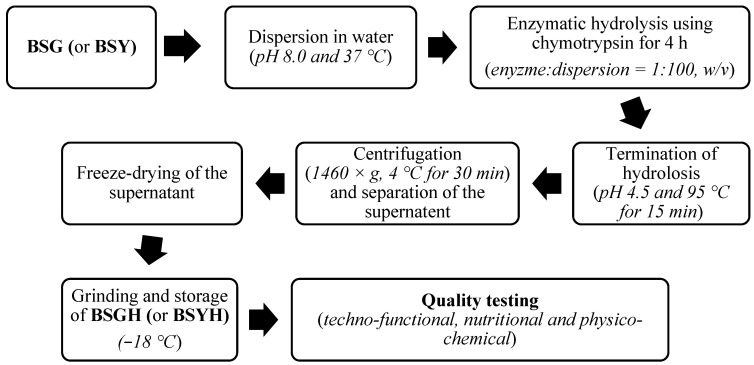
Production steps and conditions for brewer’s spent grain (BSG) protein hydrolysate (BSGH) and brewer’s spent yeast (BSY) hydrolysate (BSYH).

**Figure 2 foods-14-02882-f002:**
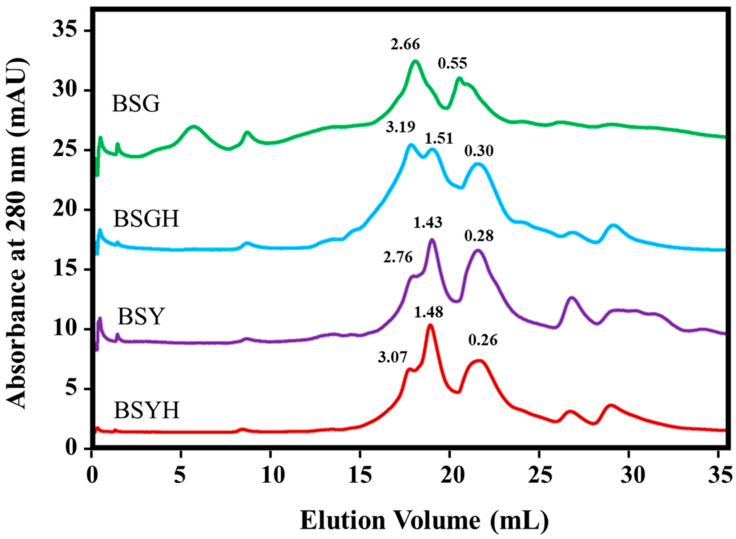
Molecular weight distribution of the peptides in the raw materials, brewer’s spent grain (BSG) and brewer’s spent yeast (BSY), and their hydrolysates (BSGH and BSYH, respectively).

**Figure 3 foods-14-02882-f003:**
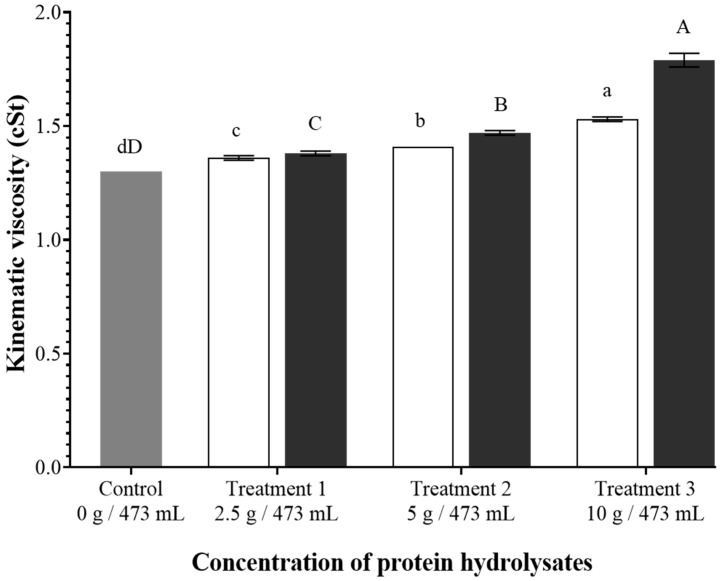
Kinematic viscosity of the beverages. Gray bar: control, white bars: brewer’s spent grain hydrolysate (BSGH) incorporated beverages, black bars: brewer’s spent yeast hydrolysate (BSYH) incorporated beverages. Lower-case and upper-case letters indicate significant differences (*p*  <  0.05) between the control and the BSGH and BSYH incorporated beverages, respectively. Error bars represent ± standard deviation, *n* = 3.

**Figure 4 foods-14-02882-f004:**
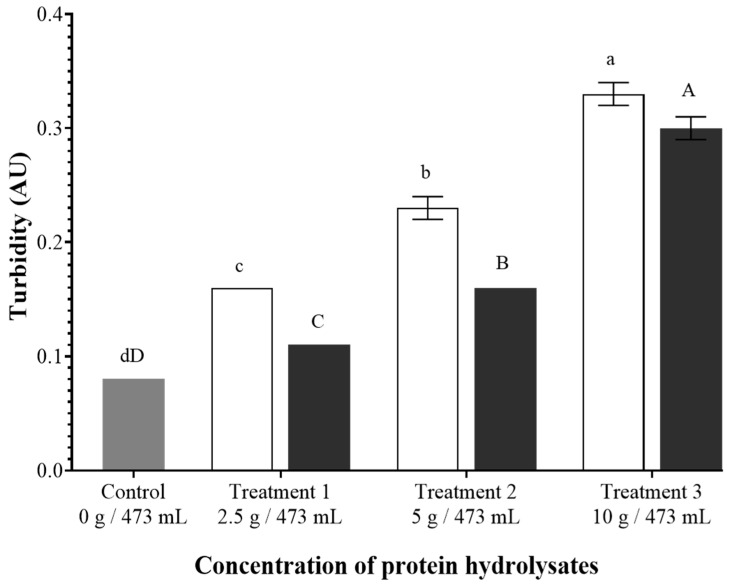
Turbidity of the beverages. Gray: control, white: brewer’s spent grain hydrolysate (BSGH) incorporated beverages, black: brewer’s spent yeast hydrolysate (BSYH) incorporated beverages. Lower-case and upper-case letters indicate significant differences (*p*  <  0.05) between the control and the BSGH and BSYH incorporated beverages, respectively. Error bars represent ± standard deviation, *n* = 3.

**Figure 5 foods-14-02882-f005:**
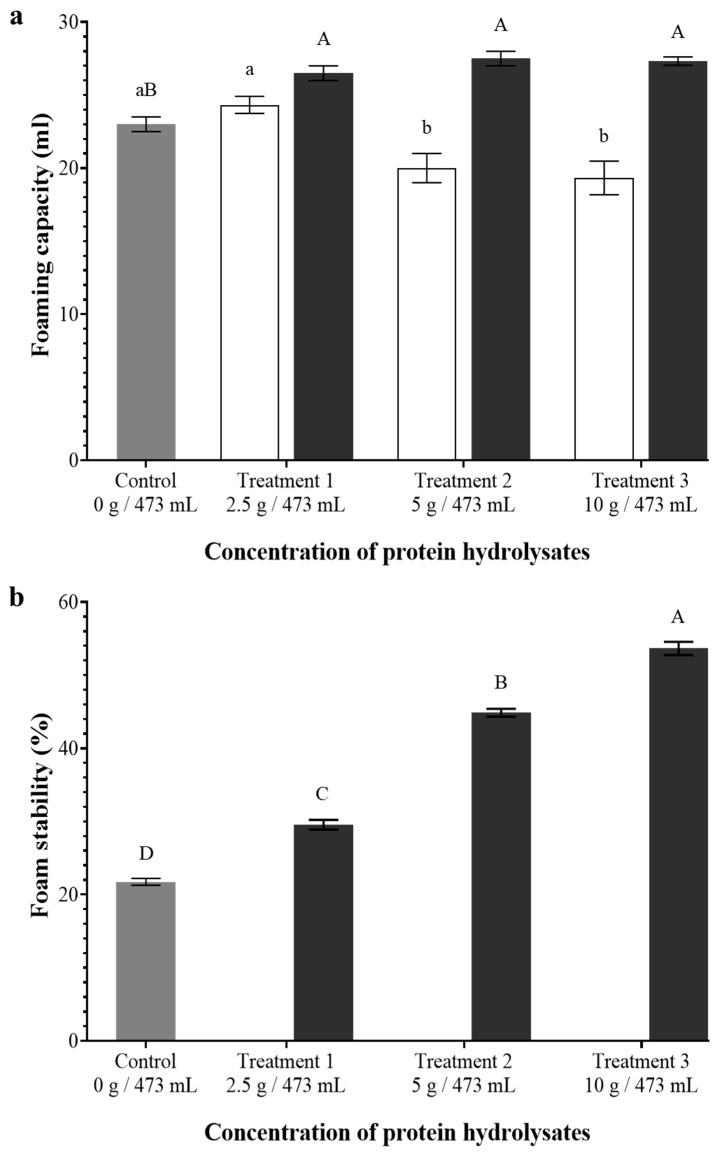
Foaming capacity (**a**) and foam stability (**b**) of the beverages. Gray: control, white: brewer’s spent grain hydrolysate (BSGH) incorporated beverages, black: brewer’s spent yeast hydrolysate (BSYH) incorporated beverages. Lowercase and uppercase letters indicate significant differences (*p*  <  0.05) between the control and the BSGH and BSYH incorporated beverages, respectively. Error bars represent ± standard deviation, *n* = 3.

**Table 1 foods-14-02882-t001:** Proximate composition of the raw materials, brewer’s spent grain (BSG) and brewer’s spent yeast (BSY), and their hydrolysates (BSGH and BSYH, respectively).

Composition	BSG	BSY	BSGH	BSYH
Moisture (g/100 g)	73.36 ± 0.23	91.14 ± 1.71	8.81 ± 0.01	13.25 ± 0.09
Protein (g/100 g db)	23.41 ± 0.18	43.59 ± 0.07	50.69 ± 0.45	50.11 ± 0.11
Fat (g/100 g db)	10.10 ± 0.23	2.86 ± 0.10	n.d.	n.d.
Ash (g/100 g db)	3.44 ± 0.14	4.4 ± 0.01	8.53 ± 0.03	14.35 ± 0.04
Total carbohydrates (g/100 g db)	62.94 ± 0.21	49.13 ± 0.59	40.78 ± 0.46	35.54 ± 0.14

Values are presented as mean ± standard deviation, *n* = 3; n.d.: not detected.

**Table 2 foods-14-02882-t002:** Techno-functional properties of the raw materials, brewer’s spent grain (BSG) and brewer’s spent yeast (BSY), and their hydrolysates (BSGH and BSYH, respectively).

Samples	WS (%)	WBC (%)	EC (%)
BSG	12.06 ± 0.38	359.44 ± 3.12	2.12 ± 0.07
BSY	17.57 ± 0.42	308.77 ± 4.11	14.95 ± 0.51
BSGH	97.98 ± 0.15	14.01 ± 0.34	2.63 ± 0.00
BSYH	96.81 ± 0.17	61.70 ± 0.82	16.28 ± 017

Values are presented as mean ± standard deviation n, *n* = 3.

**Table 3 foods-14-02882-t003:** Amino acid profiles of the raw materials, brewer’s spent grain (BSG) and brewer’s spent yeast (BSY), and their hydrolysates (BSGH and BSYH, respectively). Amin acid concentration is in g/100 g based on hydrated molecular weight.

Amino Acids (AA)	BSG	BSY	BSGH	BSYH
**Essential AA (%)**				
Histidine	1.99 ± 0.16	2.21 ± 0.06	1.87 ± 0.03	2.19 ± 0.03
Threonine	3.49 ± 0.16	5.48 ± 0.00	4.02 ± 0.05	4.78 ± 0.03
Lysine	4.59 ± 0.18	6.65 ± 0.09	4.34 ± 0.03	6.03 ± 0.04
Valine	5.36 ± 0.27	5.87 ± 0.01	5.56 ± 0.06	5.98 ± 0.04
Leucine	7.82 ± 0.41	7.36 ± 0.02	7.77 ± 0.01	7.36 ± 0.03
Isoleucine	4.24 ± 0.25	4.73 ± 0.04	4.16 ± 0.03	4.34 ± 004
Tyrosine	3.46 ± 0.24	3.97 ± 0.12	3.62 ± 0.02	4.03 ± 0.02
Phenylalanine	5.78 ± 0.35	4.70 ± 0.00	5.62 ± 0.04	4.57 ± 0.01
Cysteine	2.19 ± 0.01	2.30 ± 0.00	2.15 ± 0.01	2.86 ± 0.08
Methionine	1.84 ± 0.02	1.83 ± 0.02	1.53 ± 0.07	1.58 ± 0.03
Tryptophan	1.57 ± 0.03	1.66 ± 0.01	1.83 ± 0.00	1.97 ± 0.02
**Total**	**42.32**	**46.76**	**42.47**	**45.69**
**Non-essential AA (%)**				
Serine	4.30 ± 0.23	6.22 ± 0.04	4.74 ± 0.01	5.06 ± 0.04
Arginine	4.85 ± 0.36	4.34 ± 0.07	4.26 ± 0.03	4.07 ± 0.01
Glycine	3.79 ± 0.19	4.50 ± 0.02	3.58 ± 0.04	4.82 ± 0.04
Asx *	7.01 ± 0.25	10.02 ± 0.06	7.22 ± 0.01	10.18 ± 0.12
Glx *	22.40 ± 1.14	15.22 ± 0.01	22.55 ± 0.06	16.42 ± 0.18
Alanine	4.58 ± 0.18	6.12 ± 0.02	4.54 ± 0.01	5.86 ± 0.05
Proline	10.74 ± 0.67	6.82 ± 0.02	10.64 ± 0.06	7.89 ± 0.07
**Total**	**57.68**	**53.24**	**57.53**	**54.31**
Acidic AAs *****	29.42	25.24	29.77	26.60
Basic AAs **	11.42	13.20	10.48	12.28
Acidic/Basic AAs	2.57	1.91	2.84	2.17

* Asx = aspartic acid + asparagine; Glx = glutamic acid + glutamine; ** Histidine, lysine, and arginine. Values are presented as mean ± standard deviation, *n* = 2.

**Table 4 foods-14-02882-t004:** Color properties of the raw materials, brewer’s spent grain (BSG) and brewer’s spent yeast (BSY), and their hydrolysates (BSGH and BSYH, respectively) measured by CIE (L*, a*, b*) method.

	CIELab					
	**L***	**a***	**b***	**C**	**h**	**ΔE**
**BSG**	63.7 ± 0.3	4.3 ± 0.1	19.7 ± 0.3	20.1 ± 0.3	77.6 ± 0.1	-
**BSGH**	72.9 ± 0.3	4.4 ± 0.2	17.8 ± 0.6	18.3 ± 0.6	76.1 ± 0.1	9.2 ± 0.3
**BSY**	73.5 ± 0.4	4.5 ± 0.1	19.0 ± 0.3	19.5 ±0.3	76.8 ± 0.2	-
**BSYH**	60.8 ± 0.4	7.1 ± 0.3	22.3 ± 0.2	23.4 ± 0.3	72.2 ± 0.5	13.4 ± 0.5

Values are presented as mean ± standard deviation, *n* = 3.

**Table 5 foods-14-02882-t005:** Color properties of the brewer’s spent grain hydrolysate (BSGH) and brewer’s spent yeast hydrolysate (BSYH) incorporated non-alcoholic malt beverages, measured by ASBC and CIE (L*, a*, b*) methods.

	ASBC	CIELab					
		**L***	**a***	**b***	**C**	**h**	**ΔE**
**Control beverage**							
**Control** 0 g/473 mL	7.7 ± 0.1 ^dD^	69.6 ± 0.2 ^aA^	−0.6 ± 0.0 ^dD^	22.4 ± 0.2 ^dD^	22.4 ± 0.2 ^dD^	91.5 ± 0.1 ^aA^	-
**BSGH incorporated beverages**							
**Treatment 1** 2.5 g/473 mL	10.5 ± 0.3 ^c^	67.6 ± 0.2 ^b^	0.2 ± 0.1 ^c^	25.4 ± 0.3 ^c^	25.4 ± 0.3 ^c^	89.6 ± 0.2 ^b^	3.7 ± 0.3 ^c^
**Treatment 2** 5 g/473 mL	13.0 ± 0.1 ^b^	65.6 ± 0.1 ^c^	1.1 ± 0.0 ^b^	28.2 ± 0.2 ^b^	28.2 ± 0.2 ^b^	87.7 ± 0.0 ^c^	7.2 ± 0.2 ^b^
**Treatment 3** 10 g/473 mL	17.0 ± 0.3 ^a^	62.5 ± 0.2 ^d^	2.7 ± 0.1 ^a^	32.4 ± 0.3 ^a^	32.5 ± 0.3 ^a^	85.2 ± 0.1 ^d^	12.7 ± 0.4 ^a^
**BSYH incorporated beverages**							
**Treatment 1** 2.5 g/473 mL	9.9 ± 0.1 ^C^	67.6 ± 0.1 ^B^	0.4 ± 0.0 ^C^	26.7 ± 0.2 ^C^	26.7 ± 0.2 ^C^	89.2 ± 0.1 ^B^	4.8 ± 0.2 ^C^
**Treatment 2** 5 g/473 mL	12.6 ± 0.1 ^B^	65.4 ± 0.1 ^C^	1.6 ± 0.1 ^B^	30.9 ± 0.2 ^B^	30.9 ± 0.2 ^B^	87.0 ± 0.1 ^C^	9.7 ± 0.3 ^B^
**Treatment 3** 10 g/473 mL	18.4 ± 0.6 ^A^	61.4 ± 0.2 ^D^	4.2 ± 0.2 ^A^	37.1 ± 0.6 ^A^	37.3 ± 0.6 ^A^	83.6 ± 0.2 ^D^	17.5 ± 0.6 ^A^

Lower-case and upper-case letters indicate significant differences (*p*  <  0.05) between the control beverage and the BSGH and BSYH incorporated beverages, respectively. Values are presented as mean ± standard deviation, *n* = 3.

## Data Availability

The original contributions presented in this study are included in the article. Further inquiries can be directed to the corresponding authors.

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
