# Peer review of "Adding Value to Brewery Industry By-Products as Novel Ingredients in Non-Alcoholic Malt Beverage Applications"

_foods, 2025, doi:10.3390/foods14162882_

Round 1

Reviewer 1 Report

Comments and Suggestions for Authors

 The article submitted to the journal, entitled 'Adding Value to Brewery Industry By-Products as Novel Ingredients in Beverage Applications ’, presents a laboratory study focusing on the reuse of the two main by-products of the brewing industry for the production of a non-alcoholic beverage.

The manuscript is well structured and written in language understandable to potential readers. The methodology fits to the research objectives, except for the reservations below.  Most of the results obtained have been processed using statistical methods and are discussed. However, in my opinion, in this type of work it is necessary to explain the beverage under study. In addition, a sensory study is crucial to assess consumer acceptance of a beverage.  I therefore recommend  major revision.

Author Response

The article submitted to the journal, entitled 'Adding Value to Brewery Industry By-Products as Novel Ingredients in Beverage Applications’, presents a laboratory study focusing on the reuse of the two main by-products of the brewing industry for the production of a non-alcoholic beverage.

The manuscript is well structured and written in language understandable to potential readers. The methodology fits to the research objectives, except for the reservations below.  Most of the results obtained have been processed using statistical methods and are discussed. However, in my opinion, in this type of work it is necessary to explain the beverage under study. In addition, a sensory study is crucial to assess consumer acceptance of a beverage.  I therefore recommend major revision.

We thank the reviewer for the valuable feedback. As this study focuses on the impact of BSG and BSY hydrolysates on the physicochemical properties of a non-alcoholic beverage, the formulation used is based on a standard non-alcohol malt beverage, where the base composition remains largely consistent to alcohol free beer. We appreciate the suggestion regarding sensory evaluation and have acknowledged its relevance for future investigations in the revised discussion. Please see the changes in lines 489-492.

Comments and Suggestions for Authors

Comment 1:

Dear Authors, The topic of the manuscript "Adding value to brewing industry by-products as novel ingredients in beverages applications" is very interesting and a lot of experimental work was done. However, there are a lot of things that should be clarified or corrected in order to improve the manuscript quality.

Response 1:

We thank the reviewer for their encouraging and constructive feedback. We have carefully addressed the suggested points to improve the clarity and overall quality of the manuscript.

TITLE

Comment 2:

According to me the title should be changed. As it appears, it is expected that the use of these by-products in different beverages will be analysed, however, only one beverage has been studied in the work and, moreover, it is not known what type of beverage it is.

Response 2:

We thank the reviewer for the thoughtful comment. The title was intended to reflect the broader application potential of the by-products, though the current study focuses on non-alcoholic malt beverage, which is a standard alcohol-free beer, where only the alcohol has been removed, and the base composition remains largely unchanged. To more accurately reflect the scope of the work, the title has been revised to “Adding Value to Brewery Industry By-Products as Novel Ingredients in Non-alcoholic Malt Beverage Applications”. Please see the changes in lines 2-3.

Comment 3:

Line 33-35: this statement cannot be extracted from the study. In my opinion, it should be deleted.

Response 3:

Thank you for the observation. We agree that the original sentence may appear broad. To better align with the scope of our study, we have revised the sentence in lines 33-35 as follows:

“These findings highlight the potential of brewery by-products, after enzymatic hydrolysis, as protein-rich ingredients that can support more sustainable food systems and contribute to the nutritional enhancement of various low-protein food and beverage products.” We believe this revision addresses the concern while preserving the intended message.

INTRODUCTION

Comment 4:

The initial step in the recovery of proteins and generation of peptides from SY consists in disrupting the cell walls to release the intracellular components. The disruption of the cell walls and further protein release can be achieved by using individual or various combinations of physical, chemical or enzymatic treatments. Why has enzymatic hydrolysis been chosen?

Response 4: Enzymatic hydrolysis provides for the most time- and cost-effective release of bioactive peptides. Addition of physical and chemical treatments could provide slight increases but may not be cost-effective.

Comment 5:

When using enzymatic hydrolysis, the characteristics of the peptides obtained often depend on the protein chain from which they are released and the type of enzyme used for hydrolysis. What type of enzyme is used in this study? Why were these enzymes used?

Response 5:

We thank the reviewer for this relevant question. In this study, chymotrypsin was used as the hydrolysing enzyme. Chymotrypsin is a serine protease known for its specificity in cleaving peptide bonds at aromatic amino acids, which can lead to the generation of bioactive peptides with improved functional properties. It was selected based on preliminary trials showing that chymotrypsin was the most effective enzyme for the release of peptides from the spent grains and yeasts.

Comment 6:

In addition, it is recommended that the bibliography be updated.

Response 6:

Thank you for the recommendation. The bibliography has been updated in the revised manuscript to include more recent and relevant references.

MATERIALS AND METHODS

Comment 7:

The beverage used to carry out the study and its composition should be specified.

Response 7:

This study focuses on a non-alcoholic beverage, which is a standard non-alcoholic malt beverage , and the base composition remains largely the same among other alcohol-free beers. The percent composition of used beverage was provided in lines 115-117 as Composition %: Carbohydrate 4.2, Protein 0.1, Fat 0.0, Fiber 0.0.

Comment 8:

Line 124: remove the final ‘s’ from ‘contents’.

Response 8:

Thank you for pointing that out. The correction has been made in the revised manuscript (line 122).

Comment 9:

Line 125: the reference indicated is very old. It should be updated, if possible.

Response 9:

Thank you for the comment. The cited reference is a standard and widely accepted method for protein determination. Although it is older, it remains relevant and is still commonly used in current research and official protocols.

Comment 10:

The degree of hydrolysis is an essential parameter to indicate the level of hydrolysis of proteins to obtain peptides of different sizes and amino acid sequences. This parameter depends on several factors, including hydrolysis time. Why was a time of four hours chosen?

Response 10:

We thank the reviewer for this valuable comment. A hydrolysis time of four hours was selected based on preliminary trials aimed at balancing peptide yield, functional properties, and enzyme efficiency. Shorter durations resulted in limited hydrolysis, while extending beyond four hours showed minimal additional improvement in solubility and functional attributes. Therefore, four hours was considered optimal for obtaining peptides of desirable characteristics.

Comment 11:

This section does not indicate that the colour of the by-products used or of the protein hydrolysates has been analysed.

Response 11:

We thank the reviewer for the helpful comment. The colour analysis of both the raw by-products and their corresponding protein hydrolysates has now been reported in the Materials and Methods section of the revised manuscript (lines 219-220).

RESULTS AND DISCUSSION

Comment 12:

Line 285: the reference indicated is very old. It should be updated, if possible.

Response 12:

Thank you for the suggestion. The reference has been updated in the revised manuscript and is now cited as Reference 41 (line 285 and lines 604-606).

  1. Zhang, Yinyin, Daojing Zhang, Yajun Huang, Li Chen, Pengqi Bao, Hongmei Fang, Baocai Xu, and Cunliu Zhou. "Effects of Basic Amino Acid on the Tenderness, Water Binding Capacity and Texture of Cooked Marinated Chicken Breast." LWT 129 (2020): 109524.

Comment 13:

Table3: Statistical analysis should be included in order to evaluate statistically significant differences.

Response 13:

Thank you for the suggestion. As Table 3 is based on two replicates per sample (n=2), statistical analysis was not performed to avoid unreliable conclusions. Although we have presented the data descriptively, considering additional replicates in future studies will be important for proper statistical evaluation.

Comment 14:

Table 4: It is advisable to separate into two tables to be able to see the results better. One for by-products and protein hydrolysates and one for beverage samples.

Response 14:

Thank you for the helpful suggestion. As advised, the original Table 4 has been divided into two separate tables in the revised manuscript (lines 393-395 and 396-400). Table 4 now presents the results for raw by-products and their hydrolysates, while Table 5 presents the data for the beverage samples

Comment 15:

Figure 5b: the graph for the BSGH beverage is missing.

Response 15:

Thank you for the observation. As already stated in the manuscript, the beverages containing BSGH did not form foams with measurable stability during the 30 min measurement period at room temperature; therefore, these results are not presented in Figure 5b.

Comment 16:

The authors of the study have carried out the physicochemical, nutritional and functional characterisation of the protein hydrolysates of the two main by-products obtained in the brewing industry. Subsequently, they have added them to a beverage in different concentrations and evaluated the characteristics of these beverages. But, based on the results obtained, what is the most appropriate concentration of protein hydrolysate to use?

Response 16:

We thank the reviewer for the thoughtful comment. The purpose of this study was to determine the effect of adding BSGH and BSYH on the physicochemical properties of the beverage at different concentrations. While the results provide useful insights, identifying the most appropriate concentration for consumer use would require sensory evaluation, which is essential to assess overall acceptability, which is an important direction for future research.

Comment 17:

In addition, it will be good to make a sensory evaluation of the beverage prepared with these hydrolysates to see if it is really feasible to use these by-products of the brewing industry for this purpose. It is good to reuse by-products from other industries, but if the beverage obtained does not show good sensory characteristics, nobody will produce it.

Response 17:

We thank the reviewer for this valuable observation. We fully agree that sensory evaluation is essential to determine the practical feasibility of using these hydrolysates in beverage applications. While the present study focused on physicochemical characterization, we have acknowledged the importance of sensory assessment and highlighted it as a necessary step in future research to support potential industry application.

CONCLUSIONS

Comment 18:

Line 477: the word ‘namely’ should be deleted.

Response 18:

Thank you for the observation. The word "namely" has been removed in the revised manuscript (line 479).

Comment 19:

In the study only these hydrolysates have been added to one beverage, which is not known, therefore, the results obtained cannot be generalised to ‘non-alcoholic beverages’.

Response 19:

We thank the reviewer for this observation. In the current study, the term "non-alcoholic beverage" refers specifically to a standard non- alcohol malt beverage, and correction was made through entire paper. While the findings are based on this single formulation, the use of the broader term reflects the potential applicability of BSGH and BSYH in similar beverage systems.

Comment 20:

The results obtained show that the use of these hydrolysates causes some variations in the characteristics of the beverage used. The study should be completed to assess the acceptability of this new product by the consumer.

Response 20:

We thank the reviewer for the insightful comment. We agree that consumer acceptability is a crucial aspect. While this study focused on evaluating the physicochemical effects of BSGH and BSYH, we have acknowledged the importance of sensory analysis and suggested it as a key direction for future research in the conclusion part of manuscript.

REFERENCES

Comment 21:

References must align with the standards established by the journal. It would also be recommended to update the bibliography used to include the latest developments on the topic.

Response 21:

Thank you for the helpful comment. The references have been updated to include recent developments, and their formatting has been revised to align with the journal’s standards.

Reviewer 2 Report

Comments and Suggestions for Authors

In this paper, the authors aimed to transform BSG and BSY into protein hydrolysates through enzymatic hydrolysis and applied for non-alcoholic beverages. This topic is interesting. The research addresses a relevant industrial challenge. Overall, the data collections, the figures and table, the results and discussions were reasonable. The references are appropriate. I would like to recommend minor revisions. The detailed comments are as follows:

  1. The choice of bovine chymotrypsin enzyme is unusual for food-byproduct applications since it is expensive. Why chose it? Please explain.
  2. Degree of hydrolysis is an important index. Why missing it? Please explain.
  3. Heating at 60°C could denature proteins. Discuss potential impacts on functionality.
  4. Some error bars are missing in Fig. 4. Please add.
  5. Line 152. 2.3. is wrong. Should be 2.4.
  6. Preparation of beverages. Why chose wet basis?
  7. Check reference styles. For example, no volume and page numbers in Line 576.

Author Response

In this paper, the authors aimed to transform BSG and BSY into protein hydrolysates through enzymatic hydrolysis and applied for non-alcoholic beverages. This topic is interesting. The research addresses a relevant industrial challenge. Overall, the data collections, the figures and table, the results and discussions were reasonable. The references are appropriate. I would like to recommend minor revisions. The detailed comments are as follows:

We sincerely thank the reviewer for the positive and encouraging feedback. We are glad to know that the topic, data presentation, and overall structure of the manuscript were found to be relevant and reasonable. We appreciate the constructive suggestions provided and have carefully addressed all comments in the revised version.

Comment 1:

The choice of bovine chymotrypsin enzyme is unusual for food-byproduct applications since it is expensive. Why chose it? Please explain.

Response 1:

Thank you for your comment. Bovine chymotrypsin was chosen as a model enzyme due to its well-defined activity and reproducibility, which allowed for controlled hydrolysis. It was also the most efficient enzyme for the release of peptides from BSG and BSY during preliminary trials. Based on our own investigation, cheap sources of chymotrypsin are now available..

Comment 2:

Degree of hydrolysis is an important index. Why missing it? Please explain.

Response 2:

We thank the reviewer for the insightful comment; however, we have provided a peptide map (Fig. 2), which is also effective in illustrating the extent of protein hydrolysis (based on peptide size). We will consider including determination of the degree of hydrolysis in future work.

Comment 3:

Heating at 60°C could denature proteins. Discuss potential impacts on functionality.

Response 3:

We thank the reviewer for the helpful comment. The temperature of 60°C was selected to optimize the activity of chymotrypsin. While partial denaturation may occur, particularly in BSY proteins, it can enhance enzyme accessibility and support efficient hydrolysis. BSG proteins are generally more heat-stable and therefore less affected.

Comment 4:

Some error bars are missing in Fig. 4. Please add.

Response 4:

Thanks for your observation. Some error bars are not visible in the Figure 4 because the measurements from triplicate samples were extremely close to each other, resulting in minimal or negligible standard deviations.

Comment 5:

Line 152. 2.3. is wrong. Should be 2.4.

Response 5:

Thanks for the correction, listing in materials and methods sections was updated (lines 151-248).

Comment 6:

Preparation of beverages. Why chose wet basis?

Response 6:

Thank you for your comment. The wet basis was selected to accurately represent the actual conditions during beverage formulation and to maintain consistency throughout the preparation and analysis process.

Comment 7:

Check reference styles. For example, no volume and page numbers in Line 576.

Response 7:

Thanks for the suggestion, particular reference has been addressed in Reference18 (lines 553-554).

  1. Kadam, Deepak, Aayushi Kadam, Filiz Koksel, and Rotimi E. Aluko. "Plant-Derived Bioactive Peptides: A Comprehensive Review." Sustainable Food Proteins 2, no. 4 (2024): 183-214.

Round 2

Reviewer 1 Report

Comments and Suggestions for Authors

According to the authors, the choice of chymotrypsin was based on preliminary trials which showed it to be the most efficient enzyme to release the peptides from the grains and yeasts used. Similarly, the authors refer to the selection of a four-hour hydrolysis time based on preliminary trials aimed at balancing peptide yield, functional properties and enzyme efficacy. In my opinion, it would be important to include these trials in the article.

My comment on what is the most appropriate concentration of protein hydrolysate to use refers to making a conclusion based on the results obtained in the analysis of beverages using different concentrations of these by-products. Obviously, there is no request to identify the most suitable concentration for consumer acceptance of the product, as the article lacks a sensory study.

Finally, references in a manuscript must be meticulously checked and formatted according to the specific guidelines of the target journal.

Author Response

Comment 1:

According to the authors, the choice of chymotrypsin was based on preliminary trials which showed it to be the most efficient enzyme to release the peptides from the grains and yeasts used. Similarly, the authors refer to the selection of a four-hour hydrolysis time based on preliminary trials aimed at balancing peptide yield, functional properties and enzyme efficacy. In my opinion, it would be important to include these trials in the article.

Response 1:

We thank the reviewer for their encouraging and constructive feedback. We have carefully addressed the suggested points to improve the clarity and overall quality of the manuscript in lines 127-133 as follows:

Preliminary analysis indicated that chymotrypsin was the most effective enzyme for releasing peptides from BSG and BSY, and a hydrolysis time of 4 hours provided an optimal balance between peptide yield, functional properties, and enzyme efficiency. Papain was also used for hydrolysis under its optimal pH and temperature conditions (pH 6.5, 60 °C) for the same duration, but it showed significantly lower peptide yield compared to chymotrypsin (data available from the corresponding authors upon request).

Comment 2:

My comment on what is the most appropriate concentration of protein hydrolysate to use refers to making a conclusion based on the results obtained in the analysis of beverages using different concentrations of these by-products. Obviously, there is no request to identify the most suitable concentration for consumer acceptance of the product, as the article lacks a sensory study.

Response 2:

We appreciate the reviewer’s request for clarification on which formulation(s), in terms of concentration and hydrolysate type, most closely resembled the control beverage. Based on our comparative analysis of physicochemical parameters, we conclude that the beverage formulation containing 2.5 g/473 mL of BSYH (Treatment 1) demonstrated the closest similarity to the control across multiple properties, including color, kinematic viscosity, and moderate turbidity. Additionally, this formulation offered enhanced foaming capacity and foam stability, surpassing the control in these functional aspects. Therefore, we consider BSYH at 2.5 g/473 mL to be the optimal formulation for non-alcoholic malt beverage fortification based on its balance of functional improvements and similarity to the control product. Please see the changes in lines 494-498.

Comment 3:

Finally, references in a manuscript must be meticulously checked and formatted according to the specific guidelines of the target journal.

Response 3:

Thank you for your valuable suggestion. We have meticulously checked and revised all references to ensure they strictly adhere to the formatting guidelines of the MDPI Foods journal.